# Catalytic Preparation of Carbon Nanotubes from Waste Polyethylene Using FeNi Bimetallic Nanocatalyst

**DOI:** 10.3390/nano10081517

**Published:** 2020-08-03

**Authors:** Kezhuo Li, Haijun Zhang, Yangfan Zheng, Gaoqian Yuan, Quanli Jia, Shaowei Zhang

**Affiliations:** 1The State Key Laboratory of Refractories and Metallurgy, Wuhan University of Science and Technology, Wuhan 430081, China; likezhuo0206@163.com (K.L.); 15671628617@163.com (Y.Z.); yuangaoqian@126.com (G.Y.); 2Henan Key Laboratory of High Temperature Functional Ceramics, Zhengzhou University, 75 Daxue Road, Zhengzhou 450052, China; jiaquanli@zzu.edu.cn; 3College of Engineering, Mathematics and Physical Sciences, University of Exeter, Exeter EX4 4QF, UK

**Keywords:** carbon nanotubes, waste polyethylene, ferric nitrate, nickel nitrate, catalytic pyrolysis

## Abstract

In this work, carbon nanotubes (CNTs) were synthesized by catalytic pyrolysis from waste polyethylene in Ar using an in-situ catalyst derived from ferric nitrate and nickel nitrate precursors. The influence factors (such as temperature, catalyst content and Fe/Ni molar ratio) on the formation of CNTs were investigated. The results showed that with the temperature increasing from 773 to 1073 K, the carbon yield gradually increased whereas the aspect (length-diameter) ratio of CNTs initially increased and then decreased. The optimal growth temperature of CNTs was 973 K. With increasing the Fe/Ni molar ratio in an FeNi bimetallic catalyst, the yield of CNTs gradually increased, whereas their aspect ratio first increased and then decreased. The optimal usage of the catalyst precursor (Fe/Ni molar ratio was 5:5) was 0.50 wt% with respect to the mass of polyethylene. In this case, the yield of CNTs reached as high as 20 wt%, and their diameter and length were respectively 20–30 nm, and a few tens of micrometers. The simple low-cost method developed in this work could be used to address the environmental concerns about plastic waste, and synthesize high value-added CNTs for a range of future applications.

## 1. Introduction

As a lightweight, waterproof, and corrosion resistant material, plastic is widely used in almost every aspect of human society, contributing considerably to the economic growth and social sustainability [1]. However, the consistent increase in the accumulation of discarded plastics causes serious pollution to the environment due to their non-biodegradable nature under ambient conditions [2,3,4]. Currently, landfill and incineration are mainly used to dispose plastic waste, but they are becoming less and less acceptable due to their many demerits, e.g., waste of land and secondary pollution [5,6]. Hence, exploring a simple, low-cost and sustainable way to recycle plastic waste to form high value-added products, though very challenging, is of technical importance. 

Carbon materials such as graphite, graphene and carbon nanotubes (CNTs) have attracted a lot of attention [7,8,9,10,11,12,13,14,15,16]. Among them, the last one, which possesses excellent properties such as low density, high tensile strength, high elastic modulus, and high thermal conductivity (>3000 W·m^−1^·K^−1^) [14,15,16], is used in the fields of composite materials [17], solar cells [18] and optical devices [19]. CNTs can be synthesized by arc discharge [20], flame synthesis [21], microwave-assisted synthesis [22], chemical vapor deposition [23], and catalytic pyrolysis [24], among which, the last one has proved promising. With this method, polyolefin, which has a high carbon content, is usually used as the carbon source because the plastic waste is mostly hydrocarbon polymers [25,26,27,28,29,30]. This will not only enable the low cost production of novel CNTs but also address the above-mentioned issues caused by plastic waste accumulation and the conventional dumping methods.

In the plastic decomposition and carbon deposition process, the catalyst used plays a crucial role, which determines the morphology and structure of CNTs [31,32,33]. Conventional catalysts for the preparation of CNTs mainly include metallic iron, cobalt, nickel, and their compounds [34,35,36]. And bimetallic and polymetallic catalysts often perform better compared with monometallic counterparts. For example, by using Ni(NO_3_)_2_·0.6H_2_O, Mg(NO_3_)_2_·6H_2_O and (NH_3_)_6_Mo_7_O_24_·0.4H_2_O as catalyst precursors, and PEG200 as a solvent, Song et al. [25] firstly prepared Ni/Mo/MgO catalyst powders by a combustion method at 923 K, then prepared straight and double-helix multi-wall carbon nanotubes (MWNTs) respectively with diameters of about 30 and 60 nm from polypropylene powder under the role of the catalysts at 1123 K. Yang et al. [37] prepared Ni-Cu/MgO and Ni/MgO catalyst powders by sol-gel method from Ni(NO_3_)_2_·6H_2_O, Mg(NO_3_)_2_·6H_2_O and Cu(NO_3_)_2_·3H_2_O catalyst precursor, then used them to synthesize CNTs via thermal chemical vapor deposition method. They found that when Ni-Cu/MgO was used as a catalyst, the yield of CNTs was significantly higher than that used Ni/MgO, indicating that the incorporation of Cu enhanced the catalytic activity of Ni. By using a similar process, Baba et al. [38] firstly prepared CoO-MoO_3_/MgO catalyst powder from Co(CH_3_COO)_2_·4H_2_O, Mg(CH_3_COO)_2_·4H_2_O and (NH_3_)_6_Mo_7_O_24_·4H_2_O, then CNTs with "bamboo-like" structure were successfully formed from ethylene by using the catalyst at 923–1043 K. The combination of Mo with Co increased the carbon yield by about 20 times. Despite these positive results, there are still some problems with the catalytic pyrolysis processes, which need to be overcome, including, e.g., (1) the preparation process of bimetallic catalyst was complex, (2) relatively expensive catalysts were used and the overall production cost was high, and (3) the yield of CNTs was relatively low.

In addition, it is well known that CNTs prepared using Ni as catalyst have the advantages of high graphitization degree and good thermal stability, while Fe as catalyst has high catalytic activity and carbon solubility. Studies have also shown that the FeNi bimetallic catalysts had higher catalytic activity for catalytic pyrolysis of plastics due to the synergistic effect among components [39,40].

In the present work, CNTs were prepared from low cost waste polyethylene using inexpensive iron nitrate and nickel nitrate as catalyst precursors via simple catalytic pyrolysis method. The influence parameters of pyrolysis temperature, catalyst content and Fe/Ni molar ratio on the formation of CNTs were investigated.

## 2. Materials and Methods 

### 2.1. Raw Materials

Powdery waste polyethylene (PE, ≥97.0%) was purchased from Shanghai Runwen Material Co. Ltd. (Shanghai, China). Commercial nickel nitrate (Ni(NO_3_)_2_·6H_2_O, analytically pure) and ferric nitrate (Fe(NO_3_)_3_·9H_2_O, analytically pure) supplied by Shanghai Sinopharm Chem. Co. Ltd. (Shanghai, China) were used as catalyst precursors. All the reagents were used directly. At the same time, it has been shown that the use of nitrate or chloride as catalyst precursors has no significant effect on the formation of CNTs in our previous work [6].

### 2.2. Sample Preparation

The process of CNTs preparation is as follows: Initially, nickel nitrate and ferric nitrate were dissolved respectively into 10 mL anhydrous ethanol. Next, they were combined in pairs in different molar ratios (the molar ratio of Fe/Ni was 10/0, 8/2, 6/4, 5/5, 4/6, 2/8 and 0/10, respectively) at room temperature (298 K) under vigorous agitation by a magnetic stirrer. The resultant solution was slowly dripped into the powdery waste polyethylene (the catalyst content was respectively 0.25, 0.50, 0.75 and 1.00 wt% of the weight of polyethylene) along the cup wall via equal volume method. Subsequently, the waste polyethylene powder loaded with different catalysts precursors was obtained after 12 h drying at room temperature. Finally, the composite powder was subjected to 2 h heat treatment at 773, 873, 973, and 1073 K (the heating rate was 5 K/min) in an alumina-tube furnace, and the shielding gas was flowing argon (99.999 vol% purity).

### 2.3. Sample Characterization

The following equation was used to calculate the yield of carbon after the pyrolysis of waste polyethylene:$$C=1-\frac{\left(m_{2}-m_{3}\right)-m_{4}}{\left(m_{1}-m_{4}\right)}\times 100\%$$
where, *C* is residual carbon rate, *m*_1_ is the weight of composite powder, *m*_2_ is the total weight of composite powder and alumina crucibles before pyrolysis, *m*_3_ is the total weight of composite powder and alumina crucibles after pyrolysis, *m*_2_–*m*_3_ is the weight loss of waste polyethylene precursor before and after pyrolysis, and *m*_4_ is the weight of adding Fe/Ni catalyst theoretically.

Phases in as-prepared samples were analyzed by X-ray powder diffraction (XRD) using a Philips X’Pert Pro diffractometer (PANalytical, Hillsboro, The Netherlands). The XRD spectra were recorded in the range from 10° to 90° (2θ) with a scanning rate of 2°/min, at 40 mA and 40 kV, using Cu*kα* radiation (*λ* = 0.1542 nm). Microstructure and phase morphologies of as-prepared samples were observed by means of a scanning electron microscope (SEM; Nova400NanoSEM, 15 kV, Amsterdam, Netherlands), a transmission electron microscope (TEM; JEM-2100UHRSTEM, 200 kV, Tokyo, Japan) with an energy dispersive spectrometer (EDS, Penta FET X-3 Si (Li)), and a high-resolution TEM (HRTEM; JEM-3010, 300 kV). N_2_ adsorption-desorption isotherms were examined on an automatic surface area and pore size analyzer (Autosorb-1, Quantachrome Instruments, Boynton Beach, FL, USA). The surface area was calculated from the adsorption branch of the isotherms using non-local density functional theory, and pore size distribution was calculated using Barrett–Joyner–Halanda model.

## 3. Results and Discussion

### 3.1. Effects of Pyrolysis Temperature on the Growth of CNTs

Figure 1 shows XRD patterns of samples obtained at 773–1073 K using 0.50 wt% Fe_50_Ni_50_ (the molar ratio of Fe/Ni was 5/5) bimetallic catalysts. At 773 K, a broad diffraction peak appeared at around 26° (2θ), indicating the formation of graphitic carbon in the sample. Furthermore, at 44.2 and 51.5°, two weak diffraction peaks were observed, respectively belonging to the (111) and (200) planes of FeNi_3_ (ICDD card: No. 03-065-3244). As the pyrolysis temperature increased to 873 K, the peak height of graphitic carbon increased evidently. With further increasing the pyrolysis temperature to 973 K, the diffraction peak (about 26°) of graphitic carbon became the sharpest, and well matched to the characteristic (002) plane of graphite (ICDD card: No. 01-075-1621), revealing the highest graphitization degree of the carbon. However, the pyrolysis temperature further increasing to 1073 K led to the peak height of the carbon decreased slightly. The above results suggested that under the test conditions, 973 K was the optimal temperature for CNTs formation.

Presented in Figure 2 are SEM images of samples whose XRD patterns are displayed in Figure 1, demonstrating significant effects of the temperature on the phase morphology. At 773 K, a relatively small number of short CNTs were generated (Figure 2a). When the pyrolysis temperature increased to 873 K, the yield and lengths of CNTs increased (Figure 2b). Upon the pyrolysis temperature further increasing to 973 K, many more CNTs with relatively small diameters of about 20 nm and lengths of a few tens of micrometers were formed (Figure 2c). However, while the pyrolysis temperature further increased to 1073 K, thicker and shorter CNTs were obtained, which might be due to the relatively high temperatures that cause the catalyst particles to aggregate (Figure 2d). The SEM observations in Figure 2 conclude that the optimal formation temperature of CNTs was at 973 K, which was in agreement with that suggested by the XRD presented in Figure 1.

### 3.2. Effects of Catalyst Amount on the Growth of CNTs

XRD patterns of the samples fired at 973 K corresponding to different amounts of Fe_50_Ni_50_ bimetallic catalysts are shown in Figure 3, indicating that the amount of catalysts had only a minor effect on the phase formation in the product. When 0.25 wt% Fe_50_Ni_50_ bimetallic catalysts was added, a broad carbon diffraction peak appeared at around 26° (2θ) and two weak diffraction peaks of FeNi_3_ were also seen. With increasing the catalyst content to 0.50 wt%, the carbon diffraction peak became sharper and narrower. With further increasing the catalyst content to 0.75 wt%, the carbon diffraction peak almost did not change. When the catalyst content was finally increased to 1.00 wt%, the carbon diffraction peak decreased significantly. Thus, it can be reasonably concluded that the optimal catalyst amount required for CNTs formation was 0.50–0.75 wt%.

Figure 4 gives SEM images of the samples whose XRD patterns are presented in Figure 3, revealing the generation of CNTs with fine diameters in the samples, and the content of Fe_50_Ni_50_ bimetallic catalysts has a great influence on CNTs yield. As evidenced by Figure 4a, an extremely few number of CNTs were detected in the sample using 0.25 wt% Fe_50_Ni_50_ bimetallic catalysts. However, as the content of the catalyst increased to 0.50 wt%, many more CNTs with smaller diameters and lengths up to a few tens of micrometers were generated (Figure 4b). With the content of Fe_50_Ni_50_ bimetallic catalysts further increasing to 0.75 wt%, the CNTs were bent and intertwined despite increase in yield (Figure 4c). When the catalyst content was finally increased to 1.00 wt%, short and thick CNTs were formed and entangled together (Figure 4d). These results demonstrated that the addition of 0.50 wt% Fe_50_Ni_50_ bimetallic catalysts was best suitable for the growth of CNTs.

### 3.3. Effects of Fe/Ni molar Ratio on the Growth of CNTs

Figure 5 presents XRD patterns of samples fired at 973 K versus the molar ratio of Fe/Ni in FeNi catalyst (0.50 wt% was used), revealing the effect of the latter on the phase composition. When iron only was added as the catalyst, graphite carbon, iron and iron carbide were present as the primary phases, and when nickel only was used as the catalyst, poorly crystalline graphitic carbon (indicated by the broad peak at 26° (2θ)), along with Ni was detected. On the other hand, when the FeNi catalyst was used, graphitic carbon (ICDD card: No. 01-075-1621) became the dominant crystalline phase, along with FeNi_3_ (ICDD card: No. 03-065-3244) and Fe_2_O_3_ (ICDD card: No. 00-001-1053). In addition, with the Fe/Ni molar ratio in the catalyst increasing from 2/8 to 8/2, the graphitic carbon initially increased but then decreased. Moreover, the highest graphitic carbon peak was seen in the case of using the catalyst with the Fe/Ni molar ratio of 5/5, implying the highest graphitization degree of the carbon. These results suggested that the Fe/Ni molar ratio of 5/5 was best suitable for the formation of CNTs.

Figure 6 shows SEM images of the samples whose XRD patterns are presented in Figure 5, demonstrating the significant influence of the Fe/Ni molar ratio on the CNTs morphology. A few short and thick CNTs were generated when using 0.50 wt% iron catalyst (Figure 6a), whereas much curved CNTs were formed and entangled together when using 0.50 wt% nickel catalyst (Figure 6b). With Fe/Ni molar ratio increasing from 2/8 to 4/6, there was no obvious formation of CNTs in the sample (Figure 6c,d). However, upon increasing the molar ratio of Fe/Ni to 5/5, a large number of CNTs with diameters of 20–30 nm and lengths of a few tens of micrometers were formed (Figure 6e). Upon further increasing the Fe/Ni molar ratio to 6/4, thinner and longer CNTs were produced and entangled together (Figure 6f). Finally, when the Fe/Ni molar ratio reached 8/2, fewer CNTs with lengths of a few micrometers and diameters of 30–40 nm were formed (Figure 6g). According to the above results, the optimal Fe/Ni molar ratio for preparation of CNTs should be 5/5, which was in agreement with that suggested by the XRD results in Figure 5.

Presented in Figure 7 are the effects of Fe/Ni molar ratio in the FeNi catalyst (in total 0.50 wt%) on the carbon yield in the product samples obtained at 973 K for 2 h, verifying that the carbon yield under the circumstance of using a bimetallic catalyst was much higher than that under the circumstance of using a single metal catalyst. The carbon yield was only about 7 % and 8 % when using 0.50 wt% single metal iron and nickel. However, when the same amount of FeNi bimetallic catalyst was used, the carbon yield increased significantly. Moreover, with the Fe/Ni molar ratio in the catalyst increasing from 2/8 to 8/2, the carbon yield increased from 18 % to 22 %. In addition, the diameter, length and carbon yield of as-prepared CNTs using FeNi bimetallic catalysts and corresponding monometallic counterpart were compared (as shown in Table 1). It shows that the quality (diameter and length) of CNTs prepared by present FeNi bimetallic catalysts in this work was better. Although the carbon yield of samples prepared in some literatures [39,40] was higher, however, it should be emphasized that the length of the CNTs was shorter and the amount of the catalyst was much higher. At the same time, the yield and quality of CNTs prepared by FeNi bimetallic catalysts in this work were superior to the corresponding monometallic one. These results indicated that the FeNi bimetallic catalysts performed much better than the single Fe or Ni catalysts in enhancing the carbon yield, and catalyzing the growth of CNTs.

### 3.4. TEM/HRTEM Characterization of CNTs

TEM, HRTEM and EDS were used to characterize the morphology and microstructure of CNTs prepared under optimal conditions (973 K/2 h and 0.50 wt% Fe_50_Ni_50_ bimetallic catalysts). As displayed in Figure 8a, CNTs exhibited a clear hollow structure, with a length of a few tens of micrometers and diameter of 20–30 nm. HRTEM further reveals that the interlayer spacing of CNTs was 0.34 nm, which was consistent with the standard graphitic interlayer spacing of the (002) plane (0.34 nm), demonstrating the high crystallinity of the CNTs (Figure 8b). Some nanoparticles of about 10 nm in size were found at the tips and inside of some CNTs (indicated by the white circles in Figure 8a); they were verified by EDS (Figure 8c) to be FeNi nanoparticles. The above results indicate that FeNi bimetallic nanoparticles were prepared under the experimental conditions and acted as a catalyst to catalyze the pyrolysis of waste polyethylene to produce CNTs. Meanwhile, Figure 8a as well as our previous work [6] proved that catalyst particles always existed inside or at the tip of CNTs.

Nitrogen adsorption-desorption measurements were carried out to determine the specific surface area and pore size distribution of CNTs prepared under the optimal experimental conditions (973 K/2 h and 0.50 wt% Fe_50_Ni_50_ bimetallic catalysts), as shown in Figure 9. The CNTs presented a typical type III isotherm when the relative pressure (P/P_0_) was between 0.1 and 1.0 (Figure 9a), and based on the BET model, the specific surface areas of CNTs prepared under the optimal experimental conditions was calculated as 139.2 m^2^·g^−1^. Furthermore, the pore size distribution was calculated using Barrett–Joyner–Halanda model (Figure 9b), it was proved that the average pore size of the CNTs prepared was about 11.7 nm.

## 4. Conclusions

CNTs with a diameter of 20–30 nm and length of a few tens of micrometers were synthesized by simple catalytic pyrolysis of inexpensive polyethylene in Ar using cheap ferric nitrate and nickel nitrate as catalyst precursors. XRD, EDS and TEM confirmed that FeNi bimetallic nanoparticles were formed in-situ from the ferric nitrate and nickel nitrate catalyst precursors during the pyrolysis process, and they acted as the catalysts to produce CNTs from waste polyethylene. Compared to their single metal counterparts, the FeNi bimetallic catalysts performed much better. The optimal condition for the generation of CNTs in this work was: using 0.50 wt% Fe_50_Ni_50_ bimetallic catalysts, catalytic pyrolysis 2 h at 973 K. The work reported here could be potentially used to address the current concerns about plastic waste and to convert a range of plastic waste to high value-added CNTs.

## Figures and Tables

**Figure 1 nanomaterials-10-01517-f001:**
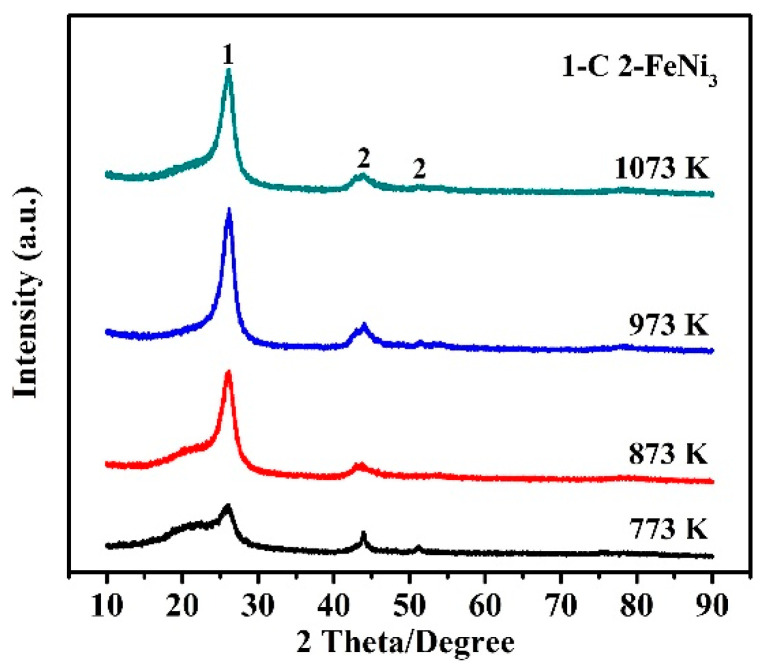
XRD patterns of samples prepared at different pyrolysis temperatures, with 0.50 wt% Fe_50_Ni_50_ bimetallic catalysts.

**Figure 2 nanomaterials-10-01517-f002:**
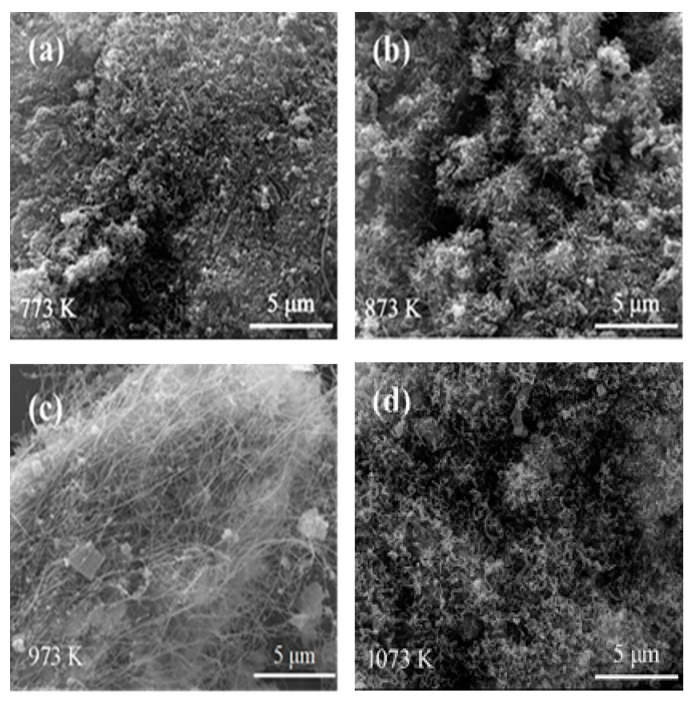
SEM images of samples obtained with 0.50 wt% Fe_50_Ni_50_ bimetallic catalysts at: (**a**) 773 K, (**b**) 873 K, (**c**) 973 K, and (**d**) 1073 K.

**Figure 3 nanomaterials-10-01517-f003:**
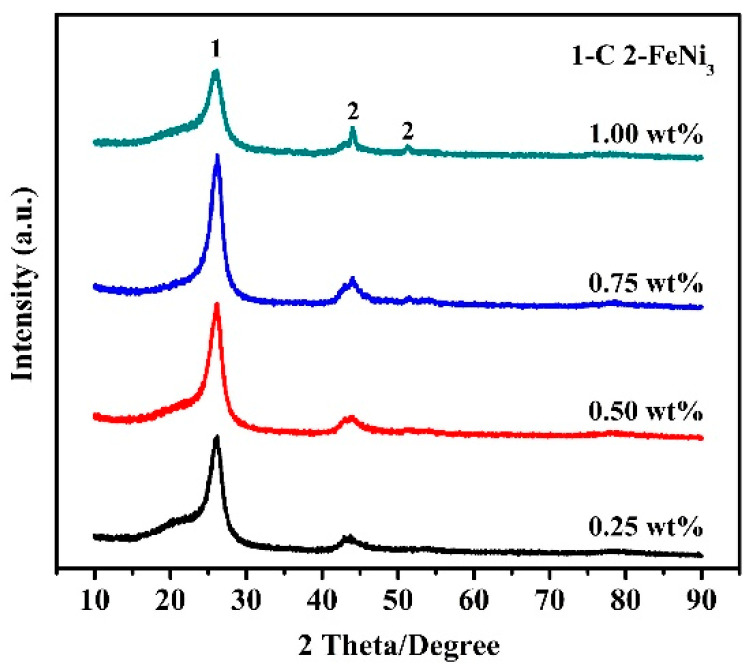
XRD patterns of samples obtained with various amounts of Fe_50_Ni_50_ bimetallic catalysts at 973 K.

**Figure 4 nanomaterials-10-01517-f004:**
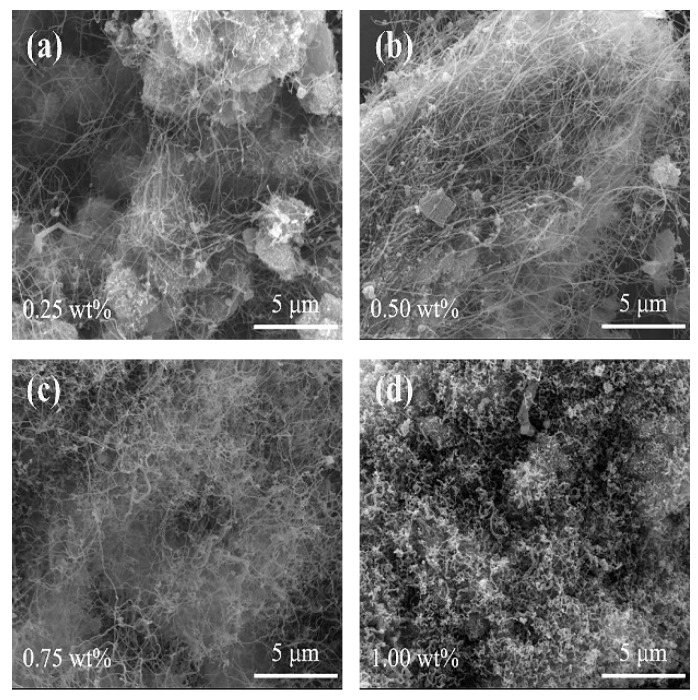
SEM images of samples obtained at 973 K with various amounts of Fe_50_Ni_50_ bimetallic catalysts: (**a**) 0.25 wt%, (**b**) 0.50 wt%, (**c**) 0.75 wt% and (**d**) 1.00 wt%.

**Figure 5 nanomaterials-10-01517-f005:**
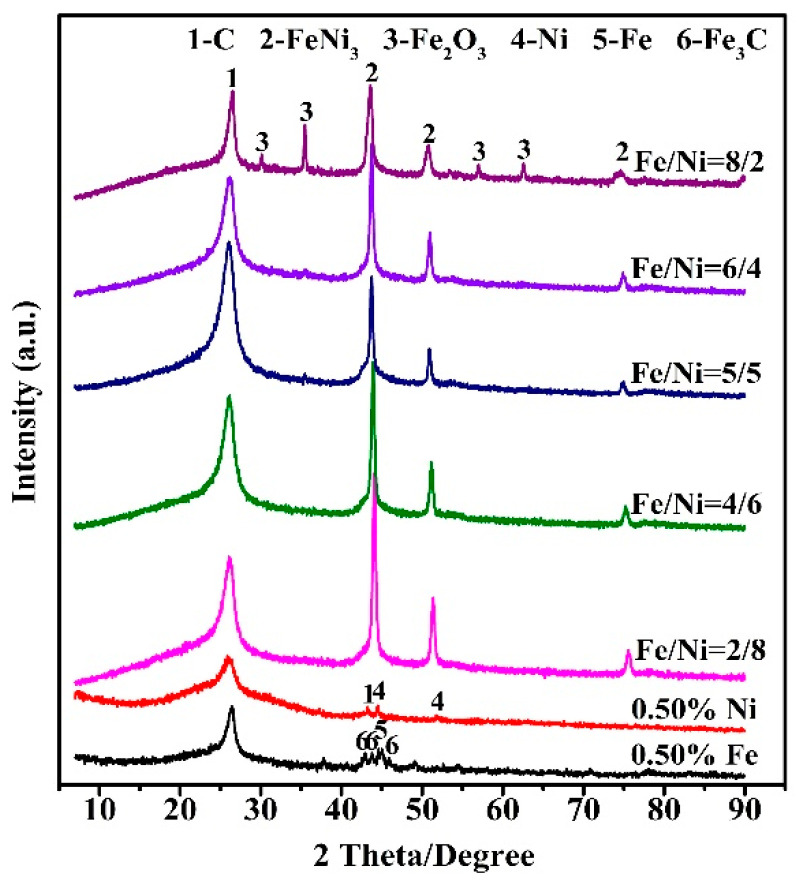
XRD patterns of product samples obtained at 973 K for 2 h, using 0.50 wt% Fe or Ni single metal catalyst and FeNi bimetallic catalyst (Fe/Ni molar ratio between 2/8 and 8/2, in total 0.50 wt%).

**Figure 6 nanomaterials-10-01517-f006:**
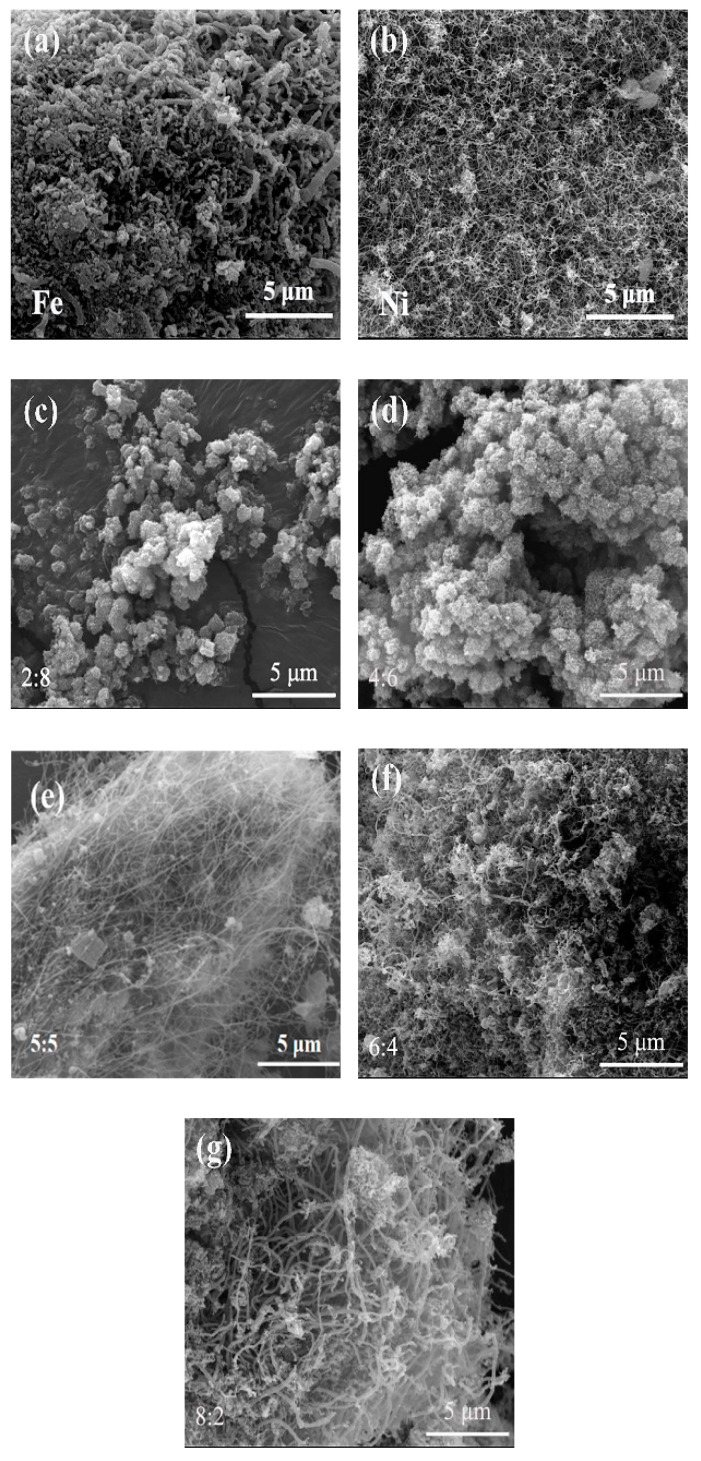
SEM images of product samples obtained at 973 K for 2 h using 0.50 wt% FeNi catalysts with different Fe/Ni molar ratio: (**a**) Fe, (**b**) Ni, (**c**) Fe/Ni = 2/8, (**d**) Fe/Ni = 4/6, (**e**) Fe/Ni = 5/5, (**f**) Fe/Ni = 6/4, and (**g**) Fe/Ni = 8/2.

**Figure 7 nanomaterials-10-01517-f007:**
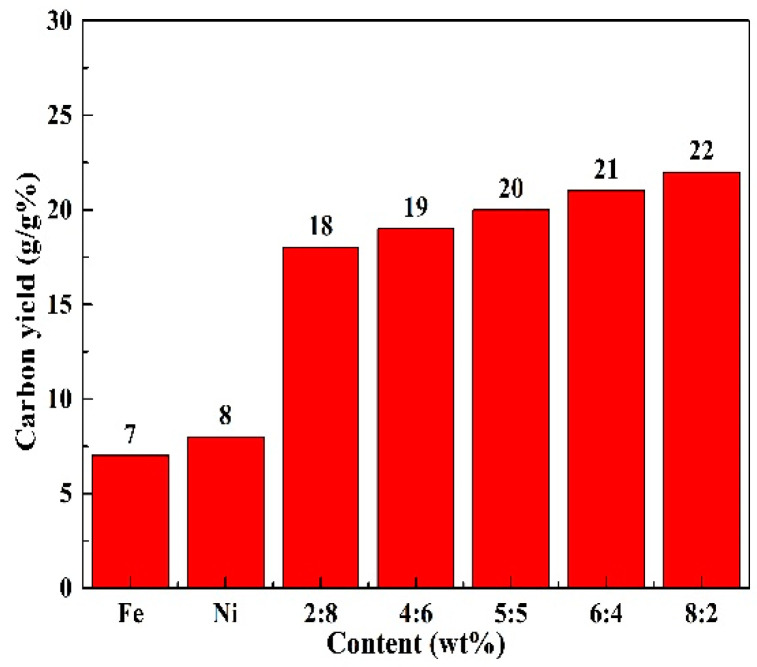
Carbon yield of product samples prepared at 973 K for 2 h, as a function of Fe/Ni molar ratio in the FeNi catalyst (in total: 0.50 wt%).

**Figure 8 nanomaterials-10-01517-f008:**
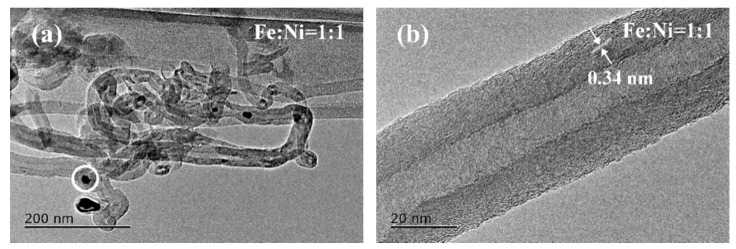
TEM and HRTEM images of product samples obtained at 973 K for 2 h using 0.50 wt% Fe_50_Ni_50_ nanocatalyst: (**a**) TEM image of CNTs, (**b**) HRTEM image of an individual CNT, and (**c**) EDS of the nanoparticles circled in Figure 8a.

**Figure 9 nanomaterials-10-01517-f009:**
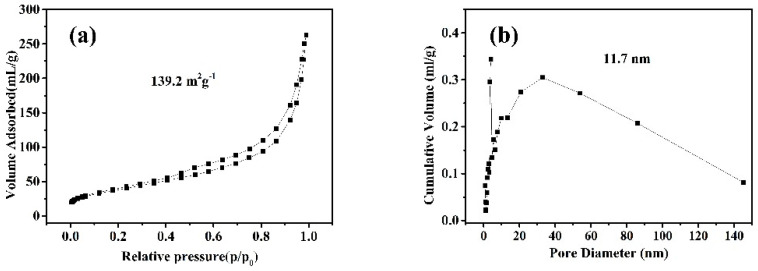
N_2_ adsorption-desorption isotherms of CNTs prepared under the optimal experimental conditions. Nitrogen adsorption/desorption isotherms (**a**) and BJH pore size distribution profiles (**b**) of as-prepared CNTs. (2 h catalytic pyrolysis at 973 K with 0.50 wt% Fe_50_Ni_50_ bimetallic catalysts).

**Table 1 nanomaterials-10-01517-t001:** Comparison of diameter, length and carbon yield of as-prepared CNTs using FeNi bimetallic and corresponding single metal as a catalyst.

Raw Material	Catalyst Type	Catalyst Addition (wt%)	Temperature (K)	CNTs Diameter (nm)	CNTs Length (μm)	Carbon Yield (wt%)	Reference
PE	FeNi	0.50	973	20–30	few tens	20	This work
PE	Fe	0.50	973	thick	shorter	7	This work
PE	Ni	0.50	973	thick	curved	8	This work
PE	Ni	0.75	973	30–50	tens	17	[6]
Mixing plastic	FeNi	5.00	1023	10–40	shorter	46	[39]
PP	FeNi	5.00	1023	20–50	shorter	41	[40]

PE: polyethylene; PP: polypropylene.

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
