# Peer review of "Catalytic Preparation of Carbon Nanotubes from Waste Polyethylene Using FeNi Bimetallic Nanocatalyst"

_nanomaterials, 2020, doi:10.3390/nano10081517_

Round 1

Reviewer 1 Report

In this paper, the authors synthesized carbon nanotubes (CNTs) by catalytic pyrolysis from waste polyethylene in Ar using an in-situ catalyst derived from ferric nitrate and nickel nitrate precursors.

The paper needs important corrections in order to be published on the journal.

  • First of all, the authors should extend the literature survey regarding the studies on the catalytic pyrolysis preparation of CNTs using FeNi bimetallic catalysts. At line 73 they stated “However, studies on the catalytic pyrolysis preparation of CNTs using FeNi bimetallic catalysts have been rarely reported”. This is not so true, since some papers are present in literature, such as 10.1016/j.enconman.2017.06.012.
  • The authors should check both the abstract and the manuscript, since some typing errors are present (for example at line 102 Fe/Ni is wrongly reported as Fi/Ni);
  • The authors prepared the catalysts in-situ, with the presence of the waste polyethylene. Did the authors prepare also only the different catalysts (without the waste polyethylene), following the described procedure? It could be interesting to characterize only the catalysts, for example through TPR studies or XRD, with the aim to deeply understand their characteristics and so their catalytic activity;
  • Did the authors perform some analysis related to the determination of the composition of the gases released during the pyrolysis of polyethylene? It could be interesting, since the H2 production during the pyrolysis may be a second added value;
  • The authors should compare the performance of their catalysts with some other similar catalysts present in literature;
  • Did the authors perform some TPO tests to check the thermal stability of carbon deposited on the catalysts?

Reviewer 2 Report

Authors describe the synthesis of CNTs by catalytic process using a combination of Fe and Ni. They found a particular ratio and thermal conditions where obtaining a relative small CNTs.

However, threre some important points in term of results and characterization which must be improved.

The methodology is based on the use of Fe and Ni , but which is the main reason?? This must be broadly discussed. Is there  influence between the metal salt used and the fabrication of the CNTs?? Please clarify this point on the manuscript.

How are the metals removed in the nanotubes formation?? or still are there??

In general, RAMAN analysis must be included for the CNT characterization in all cases.

Also experiments to determine the pore volumen or pore size distribution of the synthesized nanotubes must be included.

In the experiment about ratio Fe/Ni, has been considered the posibility to fe metallic formation?? Fig 5 shows a peak at 44-45º which could correspond to Fe metallic and this peak increases when the concentration of Fe increase. Could  this be affecting to the catalytic application ?? 

In Fig 8, High resolution TEm images (2-10 nm) and electro difraction pattern image have been included.

IN fig 8A  black aggregates can be observed, what is that?? could be Fe (0) nanoflowers??

Fig 8 corresponds to the ratio Fe/Ni 1:1??? Please including this info in the figure legend and the text.

Round 2

Reviewer 1 Report

The authors well answered my comments. In my opinion the paper can now be published on the journal

Reviewer 2 Report

Authors improved the manuscript including most of the comments , so I think this version is suitable for publication at this form